# Effects of Microecological Regulators on Rheumatoid Arthritis: A Systematic Review and Meta-Analysis of Randomized, Controlled Trials

**DOI:** 10.3390/nu15051102

**Published:** 2023-02-22

**Authors:** Tong Wu, Yanhong Li, Yinlan Wu, Xiuping Liang, Yu Zhou, Zehui Liao, Ji Wen, Lu Cheng, Yubin Luo, Yi Liu

**Affiliations:** 1Department of Rheumatology and Immunology, West China Hospital, Sichuan University, Chengdu 610041, China; 2Rare Diseases Center, West China Hospital, Sichuan University, Chengdu 610041, China; 3Institute of Immunology and Inflammation, Frontiers Science Center for Disease-Related Molecular Network, West China Hospital, Chengdu 610041, China; 4Department of Respiratory and Critical Care Medicine, Chengdu First People’s Hospital, Chengdu 610021, China; 5Meishan People’s Hospital, Meishan 610020, China

**Keywords:** rheumatoid arthritis, microecological regulators, probiotics, prebiotics, synbiotics, disease activity

## Abstract

In this study, the available data from published randomized, controlled trials (RCTs) of the use of intestinal microecological regulators as adjuvant therapies to relieve the disease activity of rheumatoid arthritis (RA) are systematically compared. An English literature search was performed using PubMed, Embase, Scopus, Web of Science and the Cochrane Central Registry of Controlled Trials and supplemented by hand searching reference lists. Three independent reviewers screened and assessed the quality of the studies. Among the 2355 citations identified, 12 RCTs were included. All data were pooled using a mean difference (MD) with a 95% CI. The disease activity score (DAS) showed a significant improvement following microecological regulators treatment (MD (95% CI) of −1.01 (−1.81, −0.2)). A borderline significant reduction in the health assessment questionnaire (HAQ) scores was observed (MD (95% CI) of −0.11 (−0.21, −0.02)). We also confirmed the known effects of probiotics on inflammatory parameters such as the C-reactive protein (CRP) (MD −1.78 (95% CI −2.90, −0.66)) and L-1β (MD −7.26 (95% CI −13.03, −1.50)). No significant impact on visual analogue scale (VAS) of pain and erythrocyte sedimentation rate (ESR) reduction was observed. Intestinal microecological regulators supplementation could decrease RA activity with a significant effect on DAS28, HAQ and inflammatory cytokines. Nevertheless, these findings need further confirmation in large clinical studies with greater consideration of the confounding variables of age, disease duration, and individual medication regimens.

## 1. Introduction

Rheumatoid arthritis (RA) is a systemic autoimmune disease with a chronic inflammatory process, which can cause symmetrical joint swelling, stiffening, arthralgia, and limited range of motion. Over time, progressive inflammation of the joints leads to cartilage damage, bone erosion, disability, and socioeconomic burdens [1,2].

The etiopathogenesis of RA is complex and involves the interaction between genetic and environmental factors. There are two major subtypes of RA (seropositive and seronegative) depending on the presence or absence of rheumatoid factor (RF) and anti-citrullinated protein antibodies (ACPAs). ACPAs can be found in sera many years before the onset of RA in approximately 67% of patients [2]. The most prevalent and strongly associated gene is the HLA-DRB1 (Major Histocompatibility Complex, Class II, DR Beta 1) region, with a 3-fold increased risk of RA [3]. In addition to genetic risk factors, the number of known environmental factors related to the risk of developing RA is also growing. The most recognized factor is cigarette smoking. It has been reported that smoking mainly affects seropositive RA, and has little effect on ACPA-negative RA [4]. Other factors such as obesity, silica dust, dietary factors, hormonal factors, and medication use also have an impact on RA development [5].

Notably, the interaction between environmental and genetic risk factors can result in a breach of immune tolerance and trigger autoimmune reactions. Emerging data show that mucosal surfaces represent the initial sites of autoimmune generation and implicated the microbiota as an extra-articular trigger of RA [6,7]. The microbiota is the term used to describe an ecological community of commensal, symbiotic, and pathogenic microorganisms, inhabiting human skin and mucosal sites [8]. The microbiota of the gastrointestinal tract contributes to metabolism, neuronal development, and physiology. It has already been observed that RA pathogenesis is associated with intestinal dysbiosis, which leads to certain autoimmune pathways and mechanisms, such as stimulation of antigen-presenting cells (APC) by activating toll-like receptors (TLRs) or nod-like receptors (NLRs), activating T cell differentiation, and alterations of intestinal permeability [9,10,11]. Multiple studies have confirmed that RA patients have significant different gut microbiota composition compared to healthy controls [12,13,14,15]. According to the results of these studies, *Prevotella copri*, *Collinsella*, and *Lactobacillus salivarius* were found to be more abundant in RA patients, while Bacteroides, *Faecalibacterium*, *Veillonella*, and *Haemophilus* were less abundant.

Current medication treatments based on the American College of Rheumatology (ACR) and European League Against Rheumatism (EULAR) guidelines manage RA from two perspectives: symptomatic treatment (NSAIDs: nonsteroidal anti-inflammatory drugs and GCs: glucocorticoids) and disease-modifying management (DMARDs: disease-modifying antirheumatic drugs) [16,17]. NSAIDs are used in the acute phase response to reduce pain by decreasing inflammation through inhibiting cyclooxygenase (COX), especially COX-2. However, the inhibition of prostaglandins can result in side effects such as rashes, bleeding, gastrointestinal ulceration, renal failure, etc. Some of the side effects can be avoided by using COX-2-selective NSAIDs (celecoxib, rofecoxib, valdecoxib) [5]. GCs can rapidly control disease activity; however, long-term use of GCs may lead to various side effects [18], such as bone thinning, infections, cardiovascular disease, diabetes, etc. DMARDs such as methotrexate (MTX) are regarded as first-line RA medication. However, around 20–30% of RA patients were unable to complete medical regimens for more than one year due to various negative responses. The most prevalent toxic effects of the medication are gastrointestinal toxicity, liver damage, and thrombocytopenia [19,20].

Considering the drawbacks of oral medications mentioned above, there is growing interest in exploring appropriate complementary therapies for RA. Microecological regulators, including probiotics, prebiotics, and synbiotics, could represent an alternative and complementary therapy to standard medications. Probiotics are live microorganisms that are intended to have health benefits when consumed or applied to the body [21]. Published data indicate that probiotics can decrease gut permeability, alleviate gastrointestinal distress, and influence systemic immune responses. These benefits could occur through maintaining a balance of “beneficial” and “harmful” bacteria in the gastrointestinal tract, ensuring adequate interactions between the gut microbiota and the mucosal immune cells, or production of inhibitory substances against pathogens [22,23,24]. Over the past few years, several probiotics have been shown to alleviate RA disease severity by altering gut microbiota in animal models and human subjects [25]. Prebiotics are functional food ingredients that are selectively metabolized by microorganisms both in vivo and in vitro, thereby supporting the proliferation of specific microorganisms and conferring health benefits to the host. Fructo-oligosaccharides and galacto-oligosaccharides are the two important groups of prebiotics. The effects of prebiotics on human health are mediated through their degradation products by microorganisms. As an example, fermentation of prebiotics by gut microbiota produces short-chain fatty acids (SCFAs). Several animal studies have implicated that SCFAs increased in collagen-induced arthritis (CIA) models fed prebiotics which presented alleviative inflammatory arthritis [26]. Synbiotics have both probiotic and prebiotic properties, and the proper combination of these two ingredients in a single product should be considered. Synbiotics were created to help probiotics survive in the intestines and they may improve colon implantation of probiotics, stimulate bacteria growth and modify the gut microbiota [27]. Based on the contributions to modulating gut microbial ecology mentioned above, microecological regulators could be a potential possibility for arthritis treatment.

The available data from randomized controlled trials (RCTs) are highly heterogeneous in terms of the study population, the characteristics of rheumatic diseases, the composition of supplements, and the results regarding activity scores and inflammatory markers. There are a few meta-analyses published to demonstrate the efficacy of probiotics or dietary supplementation in patients with inflammatory rheumatism; however, most of them included other inflammatory arthritis in addition to RA and mainly analyzed clinical variables. Furthermore, meta-analyses that discussed diet or dietary supplements conflated prebiotics with synbiotics, and thus the results are discordant and hard to interpret [25,28,29,30]. The meta-analysis we performed pooled only RCTs with a detailed composition of microecological regulator supplementation in RA and therefore to provide more conclusive results. This meta-analysis and systematic review summarizes and analyzes the efficacy of probiotics, prebiotics, and synbiotics supplementation in RA, to provide a reference for the clinical application of microecological regulators in RA patients.

## 2. Materials and Methods

This study was conducted in accordance with the Preferred Reporting Items for Systematic Reviews and Meta-Analyses (PRISMA) guidelines [31] (Appendix A). The study protocol was registered in PROSPERO International Prospective Register of Systematic Reviews (registration number CRD42022363172) [32].

### 2.1. Search Strategy

A comprehensive literature search was conducted to identify relevant articles from inception to November 2022 in the PubMed, EMBASE, Scopus, Web of Science, and Cochrane Central Registry of Controlled Trials databases. A manual search of the reference lists of all identified articles was carried out to find additional studies.

Original research articles were searched using the grouped search terms: ((“Ar-thritis, Rheumatoid”[Mesh] OR “Caplan Syndrome” OR “Felty Syndrome” OR “Rheumatoid Nodule” OR “Rheumatoid Vasculitis”) AND (“Lactobacilli” OR “Lacto-bacillus” OR “Bacillus” OR “Bifidobacteria” OR “Bifidobacterium” OR “Fructans” OR “Inulin” OR “Galacto-oligosaccharides” OR “Starch” OR “Fructo-oligosaccharides” OR “Prebiotic” OR “Probiotic” OR “Synbiotics”)) in PubMed, and((“Rheumatoid Ar-thritis” OR “Rheumatoid” OR “Caplan Syndrome” OR “Felty Syndrome” OR “Rheu-matoid Nodule” OR “Rheumatoid Vasculitis”) AND (“Lactobacilli” OR “Lactobacillus” OR “Bacillus” OR “Bifidobacteria” OR “Bifidobacterium” OR “Fructans” OR “Inulin” OR “Galacto-oligosaccharides” OR “Starch” OR “Fructo-oligosaccharides” OR “Prebiotic” OR “Probiotic” OR “Synbiotics”)) in other databases. This string was mod-ified to match each database.

Search results were retrieved, and duplicates were removed using EndNote X9 software for Windows. Three independent reviewers assessed the relevance of selected retrieved articles. Screening of titles and abstracts was followed by full-text screening. Disagreements were resolved by discussion and consensus between reviewers and senior researchers.

### 2.2. Eligibility Criteria

We included any open-label or blinded randomized controlled studies that evaluated the efficacy of oral supplementation with intestinal microecological regulators (prebiotics, probiotics, and synbiotics) in adult patients with an established diagnosis of RA. The control therapy could be a placebo or another diet intervention. We excluded any uncontrolled studies, case reports, case series, letters, editorial comments, theses, literature reviews, book chapters, news, or only abstracts. We also excluded papers if their data could not be extracted or if they were not written in English.

Outcomes included RA clinical disease activity indices such as the disease activity score of 28 joints (DAS28), number of tender or swollen joints (TJC and SJC), Health Assessment Questionnaire, Disability Index (HAQ), visual analog scale (VAS) for disease activity provided by the patient, VAS for pain and global health score (GH score). Laboratory markers were the C-reactive protein (CRP) level, erythrocyte sedimentation rate (ESR), and levels of inflammatory cytokines. Disagreements in the determination of the eligibility of each study were resolved by discussion and consensus.

### 2.3. Data Extraction

Data of interest were extracted using a custom Microsoft Excel Office spreadsheet. The following data were extracted for each study: publication date, journal, study design, sample size, demographic characteristics (e.g., age, sex, disease duration, inclusion criteria, treatments such as DMARDs and symptomatic medications (GCs and NSAIDs)), microecological regulator (prebiotics, probiotics, and synbiotics) formulation details, outcome measures, side effects, and adherence.

### 2.4. Quality Assessment

Three independent reviewers assessed the risk of bias using the Outcomes Cochrane Collaboration tool for assessing risk of bias [33]. Records limited to abstracts were not assessed because of the lack of information about the study design. Any disagreement between them was resolved by discussion.

### 2.5. Statistical Analysis

All relevant quantitative data were, where possible, pooled in the statistical meta-analyses. The outcomes were the variation between the inclusion and evaluation endpoints between the two groups. A narrative synthesis was carried out to describe data extracted from articles that could not be included in the meta-analyses.

A meta-analysis was performed for all outcomes using the RevMan V 5.3 software package developed by Nordic Cochrane Centre (Review Manager (computer program), V 5.3. Copenhagen, Denmark: The Nordic Cochrane Centre, the Cochrane Collaboration, 2011). P values lower than 0.05 were considered significant. Statistical heterogeneity of the selected studies was tested using the Q-test (χ2) and reported with the I^2^ statistic. Heterogeneity was considered to be significant when the χ2 test had a *p* value < 0.1 or I^2^ test value > 50%. A fixed-effects model was used to calculate the pooled mean difference (MD) or standardized mean difference (SMD). In the case of significant statistical or clinical heterogeneity, a random-effects model was applied. Publication bias was checked with Egger’s test.

## 3. Results

### 3.1. Study Selection

The literature search of different databases revealed 4696 records, and one additional study was identified manually as shown in Figure 1. Of these, 2342 reports were duplicated and excluded by the Endnote software. The titles and abstracts of the remaining 2355 reports were screened and 2332 reports were excluded after the screening. Then, 23 studies were excluded after screening because of the wrong type of article (n = 4), or outcome (n = 6). One study was omitted for overlap with another study published by the same researchers at the same time [34]. Twelve articles were finally included in the qualitative synthesis. One study was excluded from the meta-analysis for having a high risk of bias [35].

### 3.2. Study Characteristics

The characteristics of the included studies are summarized in Table 1. Three studies were about prebiotics [36,37,38], six about probiotics [39,40,41,42,43,44], and three about synbiotics in RA patients [35,45,46]. The intervention in two studies was a combination of probiotics and a high-fiber diet [35,46]; thus, we placed these studies in the category of synbiotics.

The characteristics of the individual studies are shown in Table 2. The total number of patients with RA in the included studies was 762; 219 patients were treated with prebiotics, 243 with probiotics and 143 with synbiotics. One study only evaluated Bacillus coagulants [41], three studies only evaluated Lactobacillus [40,42,44], and two studies assessed a mix of different probiotic types [39,43]. Lactobacillus is also the main strain in synbiotics supplementation. Two studies assessed a high-fiber diet combined with probiotic supplementation [35,46]. Three different types of prebiotics were included in this review: standard mixed dietary plant-derived polysaccharides (dPPs) [38], polyphenols [36,37] and fructan [45]. The duration of intervention ranged from 8 weeks to 1 year. There were no studies included that provided an active control involving another probiotic strain. The comparator was placebo except in Nenonen et al. [35], which compared an uncooked vegan diet mixed with lactobacilli versus a normal diet, and in Vadell et al. [46], which compared an anti-inflammatory diet mixed with probiotics versus a typical Swedish diet.

The main inclusion criteria were RA diagnosed according to the 1987 ACR/EULAR criteria [36,37,38,41,42,43]. Other inclusion criteria were a disease duration of more than 1 year [41,42,44], more than 6 months [39,45], and more than 2 years for Vadell et al. [46]. Treatments had to be consistently administered from 1 to 3 months prior to inclusion [37,38,40,42,46]. Eight studies referred to at least mild activity according to the DAS score [35,37,38,39,40,41,45,46], while four studies did not specifically require the minimum DAS score [36,42,43,44]. One study specified the requirement of no NSAIDs use [44] and two of no biological DMARDs use [39,44]

### 3.3. Risk of Bias Assessment

The risk of bias assessment is illustrated in Figure 2 and Figure 3. The assessment of the results indicated that the quality of the included papers ranged from low to high. All studies were double-blinded random control studies, except two single-blinded studies [35,46] and one open-label study [36]. In addition, some studies did not adequately report key outcomes, such as the DAS28 and CRP level, which were also considered to have a moderate risk [41,42,44]. Nenonen et al. [35] was rated high risk because of the inappropriate measurement and data reporting of the outcome; thus, we excluded it from the meta-analysis.

### 3.4. Outcomes

#### 3.4.1. Disease Activity Indices

DAS28

Nine studies provided data on the DAS28: four involved probiotics as an intervention [39,40,43,44], three involved prebiotics [36,37,38], and two involved synbiotics [45,46]. Our meta-analysis pooled eight RCTs with complete outcome data and revealed a significant effect in reducing the DAS28 (MD −0.47 (95% CI −0.90, −0.05) *p* = 0.03, I^2^ = 90%, n patients = 462). Subgroup analysis based on microecological regulator types further indicated that prebiotics were statistically significant at reducing the DAS28 with an MD (95% CI) of −1.01 (−1.81, −0.2) (*p* = 0.01, I^2^ = 78%; n patients = 140), while neither probiotic nor synbiotic supplementation were statistically significant (Figure 4).

TJC-28 and SJC-28

Four studies [39,40,42,44] reported the impact of probiotics on TJC-28 and SJC-28, and two provided data on prebiotics [36,38]. The MD (95% CI) [−0.55 (−0.90, −0.20)] obtained in the current study (253 subjects) of the TJC-28 was statistically significant (*p* = 0.002). The same meta-analysis was conducted on SJC-28 and showed no statistically significant correlation between microecological regulator supplementation and SJC-28 (Figure 5).

#### 3.4.2. Life Impact

HAQ

Two studies provided data on probiotics supplementation based on the HAQ [40,42], and two provided data on prebiotics supplementation [37,38]. In total, a borderline statistically significant reduction in HAQ scores was observed (MD −0.11 (95% CI −0.21, −0.02) *p* = 0.02, I^2^ = 0%, n patients = 148) (Figure 6A).

VAS of pain

Four studies provided direct data on pain in RA patients as measured with a 10 mm VAS [39,40,41,45]. We pooled all four studies in the meta-analysis, and no significant influence on the VAS of pain score was found, with an MD −0.49 (95% CI −1.40, 0.42) *p* = 0.29, I^2^ = 86%, n patients = 182) (Figure 6B).

#### 3.4.3. Inflammatory Markers

CRP

Six studies provided data on the effects of probiotics on the CRP level [39,40,41,42,43,44], two of prebiotics [36,38], and one of synbiotics [45]. We pooled seven RCTs with complete outcome data and revealed a significant effect in reducing CRP levels (MD −1.82 (95% CI −3.29, −0.35), *p* = 0.02, I^2^ = 71%, n patients = 349) [36,39,40,42,43,44,45]. In addition, subgroup analysis of probiotics indicated a negative effect size, as shown in Figure 7.

ESR

Six studies were pooled in our meta-analysis [36,37,40,42,43,46], and no significant correlation was observed between the microecological regulators and ESR (MD −3.20 (95% CI −8.65, 2.26), *p* = 0.25, I^2^ = 79%, n patients = 323) (Figure 8).

IL-1β, IL-6 and TNF-α

Only three studies of probiotics in RA patients provided data on IL-1β [40,42,44]. All three studies were pooled in the meta-analysis and demonstrated a significant improvement in the IL-1β level as a result of probiotic supplementation in 93 RA patients (MD −7.26 (95% CI −13.03, −1.50), *p* = 0.01, I^2^ = 33%).

The above studies and one study of prebiotics provided data on IL-6 and TNF-α [36,40,42,44]. We pooled the four studies and no significant IL-6 and TNF-α improvement were noticed in the meta-analysis (Figure 9).

#### 3.4.4. Tolerance Data

Seven studies provided information on side effects, and no side effects related to microecological regulators were reported in six of them [35,36,37,38,39,40,41,43,44,45]. Vadell et al. [46] reported that 29% of patients during the intervention periods experienced upset stomach symptoms, and most of them only existed at the start of the intervention period.

## 4. Discussion

The aim of this systematic review and meta-analysis was to identify the respective efficacy of intestinal microecological regulators in RA patients. Several statistically significant and possibly clinically meaningful effects were observed with microecological regulator intervention: (1) a significant decrease in the DAS28, (2) borderline benefits according to the HAQ, (3) a significant decrease in CRP levels in RA patients under probiotics, and (4) a significant decrease in proinflammatory cytokines in the probiotics group. This information may help inform clinical physicians and RA patients concerning the use of intestinal microecological regulators.

All except one study were rated with a low risk of bias arising from the randomization process for using appropriate random sequence generation and allocation concealment methods. Cannarella et al. [43] was considered to raise some concerns, because whiel they provided randomization methods, they did not report allocation concealment methods. Nine studies were rated as having a low risk of assignment to intervention bias, but Khojah et al. [36], Nenonen et al. [35], and Vadell et al. [46]. raised some concerns, because they were not double-blind RCTs. Alipour et al. [44], Mandel et al. [41], and Zamani et al. [39] were rated as raising some concerns regarding attrition bias due to outcome data loss at follow-up. Khojah et al. [36] and Mandel et al. [41] did not adequately report key outcomes, and were considered to raise some concerns regarding bias in the measurement of the outcomes, and Nenonen et al. [35] was rated as being of high risk because of the inappropriate measurement and data reporting of the outcomes. Due to the lack of a registered protocol or the insufficient data on several secondary outcomes, six studies were considered to raise some concerns regarding reporting bias. Overall, seven studies were rated as having a low risk of bias, four were rated as raising some concerns, and one was rated as having high risk. The two funnel plots of DAS28 and CRP were substantially symmetrical (Appendix A). The results of Egger’s test for DAS28 and CRP were *p* value = 0.537 and *p* value = 0.383, respectively, indicating that there was no publication bias.

We first analyzed the ability of microecological regulators to relieve the symptoms of RA patients. The present study found significant benefits of microecological regulators intervention on DAS28 reduction, which was consistent with the findings of Zamani et al. [39] and Alipour et al. [44]. In fact, we further identified a strong DAS28 response in RA patients to probiotics after removing Pineda et al. (MD −0.25 (95% CI −0.42, −0.08), *p* = 0.003). This might be because of the strict inclusion criteria, which require patients to have at least four swollen and four tender joints at enrollment. This led to a small sample size of 29 patients and perhaps a failure to demonstrate the efficacy of probiotics as an adjunctive therapy within three months. Prebiotics, more specifically, polyphenol supplementation, showed a better impact on disease activity in our subgroup analysis (Appendix A). These results were consistent with our previous knowledge that probiotics and prebiotics can alleviate joint inflammation in CIA models [47,48,49,50].

Concerning quality of life, some researchers have proposed that the use of the HAQ may better reveal the functional status of RA patients in comparison to physical examinations and laboratory indicators [40]. The limited number of included RCTs may make it difficult to specifically evaluate the quality-of-life impact, and a borderline significant improvement in HAQ scores was observed. However, no significant effect was observed in subgroup analysis. This result was consistent with a previous meta-analysis conducted by Lowe J et al. [51].

Concerning the inflammatory markers, participants subjected to microecological regulator supplementation showed a borderline significant reduction in CRP levels. However, the reduction in CRP levels (MD −1.82 (95% CI −3.29, −0.35), *p* = 0.02) may not represent a clinically meaningful change. The pooled result might be influenced by two trials that employed a more sensitive test (hs-CRP instead of CRP) and had larger sample sizes. Regarding changes in the ESR, some studies reported normal baseline values of the ESR, but they did not provide the data, and this information could not be extracted a posteriori. Moreover, levels of IL-1β showed a significant decrease in the probiotics group, which was consistent with studies conducted by Alipour et al. [44] and Khojah et al. [36].

Regretfully, due to the fairly limited number and high heterogeneity of eligible RCTs in respective subgroups, we were not able to select the most effective type of probiotics or prebiotics by comparing our present data. From the above analysis, however, we can propose that probiotics could be more effective than prebiotics for RA patients in adjunct with disease-centered treatment. In addition, in prebiotics, polyphenol supplementation is more likely to be used as adjuvant therapy. Synbiotics has no obvious advantage over the two in our meta-analysis.

Many studies have proposed a “gut–joint axis” and suggested that inflammation in the gut mucosa can precede joint manifestations [52]. The gut microbiota is related to RA etiology through several autoimmune pathways, such as the regulation of T helper and T reg cell functions and the induction of immune tolerance. Besides probiotics’ local effect on gut health, such as diminishing harmful bacteria, data from animal and human studies revealed that probiotics modulate locally and systemically the immune system [11,53,54,55]. Evidence from clinical and animal studies suggested SCFAs as possible mediators of these functions. SCFAs are also the main fermentation products of prebiotics by gut microbiota. It can influence the B lymphocyte’s cellular proliferation, inhibit germinal center B cell, and plasmablast differentiation, as well as innate natural killer T (NKT) cell cytokine production [56,57]. Furthermore, animal and human studies have shown that prebiotics can improve immunity functions by increasing the population of protective microorganisms and decreasing the population of harmful bacteria by *Lactobacilli* and *Bifidobacteria* [58,59]. Beyond SCFAs, probiotics modulate the immune response by directly affecting the immune system. Concerning the innate immune system, probiotics can blind specific TLRs, affect downstream signaling, promote the expression of proteins that negatively regulate TLRs activity, and thus reduce inflammation induced by different pathogens [23,60]. T cells are essential to the adaptive immune response. Several animal studies of RA have shown that probiotics tend to generate a Treg immune response, promote the conversion of T cells into Tregs expressing the forkhead box transcription factor (FoxP3), and enhance the suppressive function of pre-existing Tregs [61,62]. The increase in anti-inflammatory and the decrease in pro-inflammatory cytokines are both associated with the upregulation of FoxP3-positive Treg cells.

We analyzed the variation in the outcome values before and after supplementation, as Mohammed et al. [30] and Sanchez et al. [63] did, thus eliminating differences in baseline data between the intervention and control groups. The strengths of our meta-analysis are that we compared variations in outcome measures between the two groups and included only RCTs which were rated as having a low or moderate risk of bias. Another strength of our meta-analysis is that we pooled only studies that were human RCTs with a detailed composition of microecological regulators. This makes our analysis results more concordant and easier to interpret. We extensively and comprehensively analyzed the effects of different types of prebiotics, probiotics, and synbiotics supplementation in RA patients, provided the effect of pro/pre/synbiotics clearly and separately based on the stringent definition and discrimination of all these three supplementations, and showed a great variety of microecological regulator treatment options in use.

The most significant limitation of the present study was the insufficient number of RCTs that were eligible for analysis and the high heterogeneity, which affects the conclusion of our study. On the other hand, the variation in data and the presence of incomplete data can also affect the reliability and validity of the results. The sample sizes across the included studies were generally small, and therefore, the clinical significance of outcome changes was insufficient. This systematic review of the literature provided very different baseline characteristics and inclusion criteria for RA patients. As the average disease duration was approximately 9 years in all studies, the current analysis did not provide information on patients with newly diagnosed RA.

## 5. Conclusions

All of the above findings suggest that intestinal microecological regulators have great potential to improve the outcome of established therapies in RA patients, especially prebiotics in improving symptom severity and probiotics may have a promising role in upregulating inflammatory markers such as CRP and IL-1β. Nevertheless, further studies are required to consolidate these effects and further investigate the efficacy and safety of microecological regulators in newly diagnosed RA patients, particularly preclinical RA patients.

## Figures and Tables

**Figure 1 nutrients-15-01102-f001:**
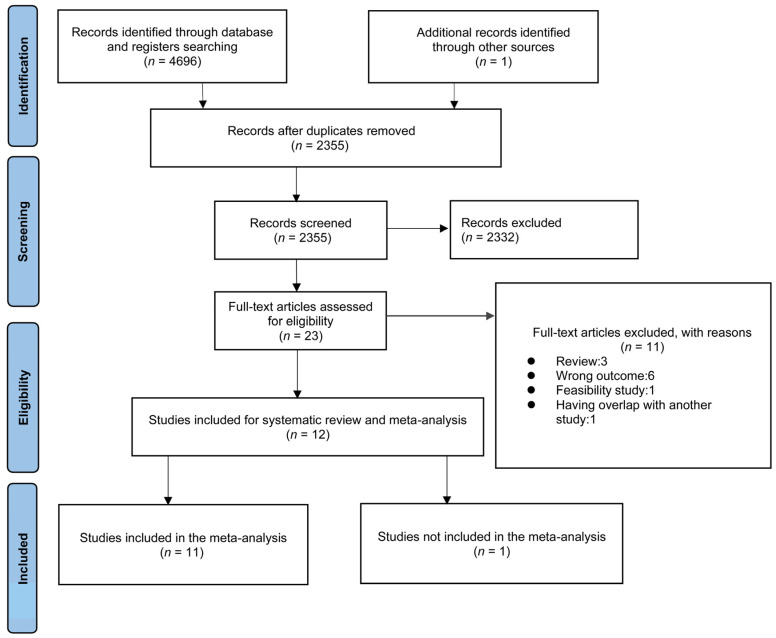
Preferred reporting items for systematic reviews and meta-analyses (PRISMA) flow diagram.

**Figure 2 nutrients-15-01102-f002:**
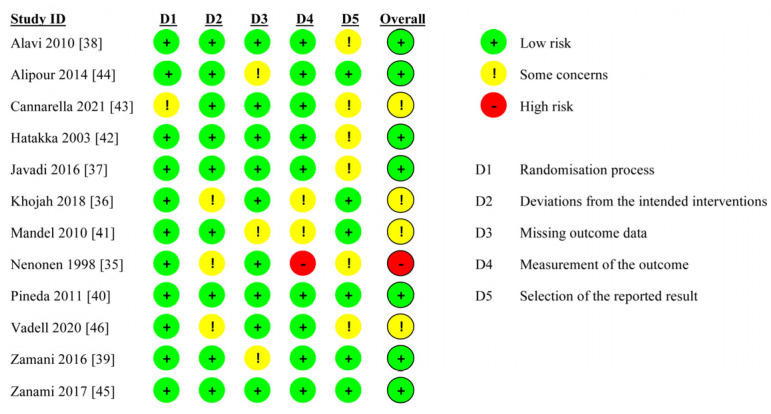
Summary of risk-of-bias judgments within each bias domain of the Cochrane Collaboration tool [35,36,37,38,39,40,41,42,43,44,45,46].

**Figure 3 nutrients-15-01102-f003:**
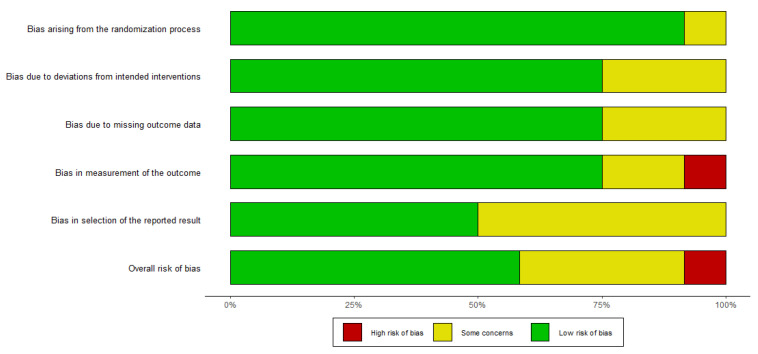
Risks of bias graph expressed as percentages.

**Figure 4 nutrients-15-01102-f004:**
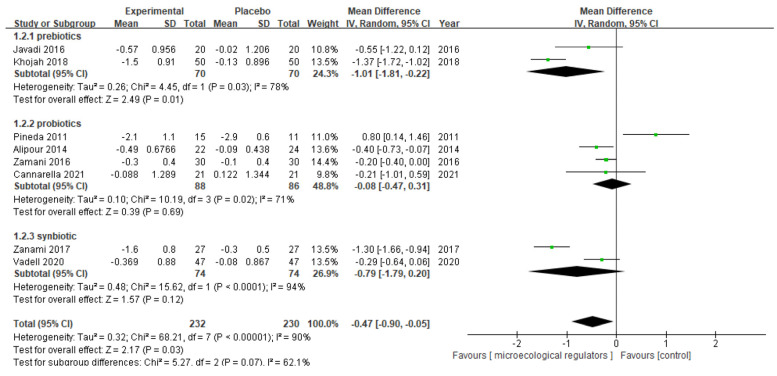
Forest plot of disease activity score (DAS28) variations in rheumatoid arthritis [36,37,39,40,43,44,45,46].

**Figure 5 nutrients-15-01102-f005:**
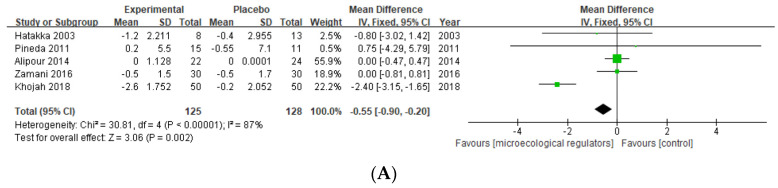
(**A**) Forest plot of TJC-28 variations in rheumatoid arthritis; (**B**) forest plot of SJC-28 variations in rheumatoid arthritis [36,39,40,42,44].

**Figure 6 nutrients-15-01102-f006:**
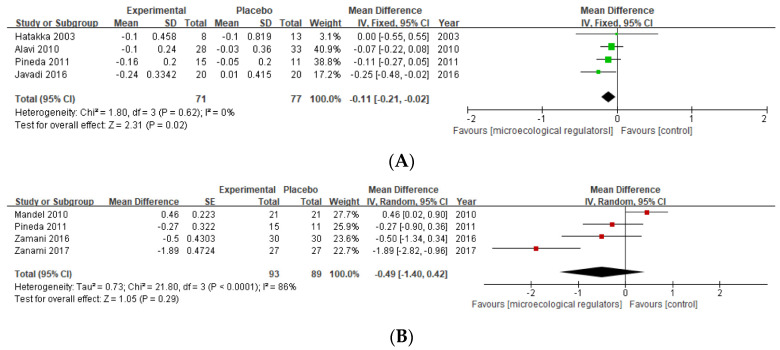
(**A**) Forest plot of HAQ variations in rheumatoid arthritis; (**B**) forest plot of pain—VAS variations in rheumatoid arthritis [37,38,39,40,41,42,45].

**Figure 7 nutrients-15-01102-f007:**
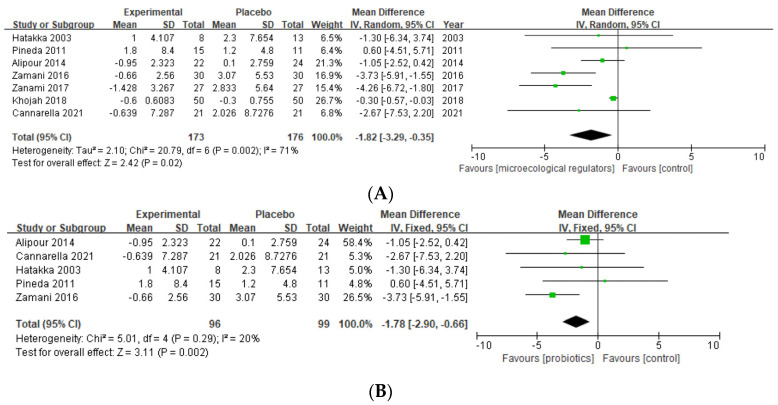
(**A**) Forest plot of CRP variations in rheumatoid arthritis; (**B**) subgroup analysis of probiotics [36,39,40,42,43,44,45].

**Figure 8 nutrients-15-01102-f008:**
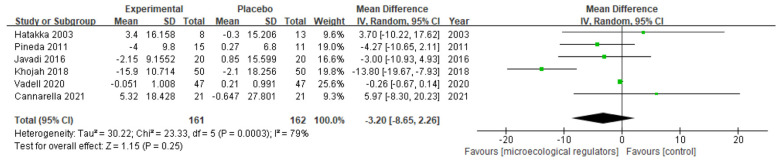
Forest plot of ESR variations in rheumatoid arthritis [36,37,40,42,43,46].

**Figure 9 nutrients-15-01102-f009:**
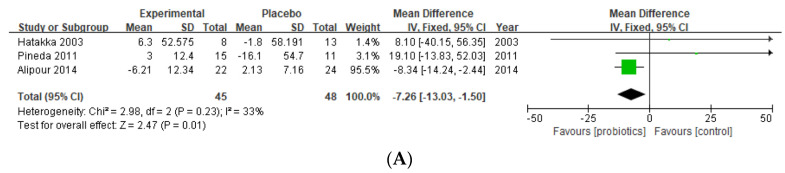
(**A**) Forest plot of IL-1β variations in rheumatoid arthritis; (**B**) forest plot of IL-6 variations in rheumatoid arthritis; (**C**) forest plot of TNF-α variations in rheumatoid arthritis [36,40,42,44].

**Table 1 nutrients-15-01102-t001:** Baseline characteristics of patients included in the RCTs.

Study	Country	Inclusion Criteria		Groups	Age (Years) Mean (SD)			Disease Duration (Years) Mean (SD)		Activity Score Mean (SD)	Current Medication		
											DMARDs N (%)	Oral CS N (%)	NSAIDs N (%)
**Prebiotics: n = 3**													
Alavi et al., 2010 [38]	UK	ACR criteria, ≥18 years, active disease, stable regime of treatment ≥ 2 months		Prebiotics	60 (10.5)			NR		DAS28 score 4.29	NR	NR	NR
				Placebo				NR					
Javadi et al., 2016 [37]	Iran	Women, ACR 1987, 19–70 years, active disease, stable regime of treatment ≥ 1 month		Prebiotics	46.55 (9.94)			5.17 (3.83)		NR	MTX 23 (92) HCQ 18 (72) SSZ 4 (16) Cyclosporine 3 (12)	Prednisolone 19 (76)	6 (24)
				Placebo	48.00 (8.39)			4.87 (3.03)		NR	MTX 23 (92) HCQ 20 (80) SSZ 2 (8) Cyclosporine 2 (8)	Prednisolone 19 (76)	8 (32)
Khojah et al., 2018 [36]	Egypt	ACR criteria		Prebiotics	46.5 (12.3)			9.4 (5.8)		DAS28-ESR 4.62 (0.99)	NR	NR	NR
				Placebo	44.2 (16.4)			9.8 (5.5)		DAS28-ESR 4.91 (0.92)			
**Probiotics: n = 6**													
Hatakka et al., 2003 [42]	Finland	ACR 1987, 18–64 years, disease duration > 1-year, stable regime of treatment ≥ 3 month		Probiotics	50 (10)		8.3 (7.3)			NR	100	6 (75)	6 (75)
				Placebo	53 (7)		11.0 (8.2)			NR	100	8 (62)	10 (77)
Mandel et al., 2010 [41]	USA	ACR 1987, 18–80 years, disease duration ≥ 1-year, oral CS < 10 mg/day, four or more among: MS ≥ 1 h, STS in ≥3 joint areas, swelling of IPP or MCP or wrist joints, rheumatoid nodules, RF+, erosions		Probiotics	62.5		NR			NR	18 (78)	NR	2 (9.1)
				Placebo							17 (77)	NR	3 (13.6)
Pineda et al., 2011 [40]	Canada	ACR criteria, 18–80 years, ≥4 swollen and tender joints, stable regime of treatment ≥ 1 month		Probiotics	63.8 (7.5)		19 (12.4)		DAS28 4.18 (1.05)		MTX 11 (73) HCQ 6 (40) Leflunomide 3 (20) SSZ 5 (33)	4 (26)	NR
				Placebo	59.1 (9.1)		13.7 (8.4)		DAS28 4.83 (0.91)		MTX 11 (78) HCQ 7 (50) Leflunomide 3 (21) SSZ 4 (28) Myochrysine 2 (14)	3 (21)	NR
Alipour et al., 2014 [44]	Iran	Woman, ACR 1987, 20–80 years, disease duration ≥ 1 year, CRP < 5.1, no NSAIDs or bDMARDs, oral CS < 10 mg/day		Probiotics	44.29 (9.77)		5.25 (3.75,10.0)		DAS28-CRP 2.56 (1.01)		MTX 15 (68.2) HCQ 18 (81.8)	Prednisolone 21 (95.5)	NR
				Placebo	41.14 (12.65)		4.75 (3.0,9.0)		DAS28-CRP 2.31 (0.90)		MTX 20 (83.3) HCQ 18 (75.0)	Prednisolone 23 (95.8)	NR
Zamani et al., 2016 [39]	Iran	ACR 1987, 25–70 years, disease duration ≥ 6 months, DAS28 > 3.2, no biological DMARDS.		Probiotics	52.2 (12.2)		7.0 (5.7)		DAS28-CRP 4.0 (0.7)		MTX 29 (96.7) HCQ 20 (66.7)	Prednisone 27 (90.0)	NR
				Placebo	50.6 (13.1)		7.0 (6.7)		DAS28-CRP 4.1 (0.7)		MTX 29 (96.7) HCQ 21 (70.0)	Prednisone 28 (93.3)	NR
Cannarella et al., 2021 [43]	Brazil	ACR 1987, ≥18 years,		Probiotics	59 (49,68)		18 (10,25)		DAS28-ESR 3.20 (2.47-4.21)		MTX 17 (80.95) HCQ 7 (33.33) Leflunomide 11 (52.38)	Prednisone 7 (33.33)	NR
				Placebo	57 (48,64)		12 (7,20)		DAS28-ESR 3.83 (2.75-4.69)		MTX 10 (47.61) HCQ 7 (33.33) Leflunomide 8 (38.09)	Prednisone 15 (71.42)	NR
Synbiotics: n = 3													
Nenonen et al., 1998 [35]	Finland	ARA criteria, Steinbrocker’s functional class II–III, SJC > 3 or TJC > 5, ESR > 20 mm/h or CRP > 10 mg/L	Probiotics+ uncooked vegan diet			49.1 (7.1)	12.6 (10.3)			DAS28-ESR 3.26	MTX 10 (52.6)	10 (52.6)	16 (84.2)
			Normal diet			55.6 (10.8)	16.1 (13.6)			DAS28-ESR 3.44	MTX 5 (25)	9 (45)	18 (90)
Zamani et al., 2017 [45]	Iran	ACR 1987, 25–70 years, disease duration ≥ 6 months, DAS28 > 3.2.	Synbiotics			49.3 (11.0)	7.7 (6.1)			DAS28-CRP 4.2 (0.7)	MTX 26 (96.3) HCQ 19 (70.4)	Prednisolone 24 (88.9)	NR
			Placebo			49.5 (12.9)	7.5 (6.4)			DAS28-CRP 3.5 (0.8)	MTX 26 (96.3)HCQ 18 (66.7)	Prednisolone25 (92.6)	NR
Vadell et al., 2020 [46]	Sweden	18–75 years, disease duration ≥ 2 years, DAS28-ESR ≥ 2.6, stable regime of DMARDs treatment ≥ 8 weeks	Probiotics + high-fiber diet			61 (12)	20.0 (9.5)			DAS28-ESR 3.8 (0.9)	MTX 31 (66) SSZ 6 (13) anti-TNF 16 (34)	12 (26)	24 (51)
			Normal diet							DAS28-CRP 3.6 (0.8)			

Age and disease duration are presented as mean and standard deviation (SD). Current medications are presented as number and percentage (%). ACR: American College of Rheumatology; ARA: American Rheumatism Association; anti-TNF: anti-tumor necrosis factor; bDMARDs: biological disease-modifying antirheumatic drugs; CS: corticosteroid; DMARDs: disease-modifying antirheumatic drugs; CRP: C-reactive protein; DAS28: disease activity score in 28 joints; ESR: erythrocyte sedimentation rate; HCQ: hydroxychloroquine, IPP: inter phalangeal proximal; MCP: metacarpophalangeal; MS: morning stiffness; MTX: methotrexate; NR: not reported; NSAIDs: non-steroidal anti-inflammatory drugs; RCT: randomized controlled trial; RF: rheumatoid factor; SD: standard deviation; SJC: swollen joint count; SSZ: sulfasalazine; STS: soft tissue swelling; TJC: tender joint count; USA: United States of America; UK: United Kingdom.

**Table 2 nutrients-15-01102-t002:** Study characteristics of the 12 studies included in the systematic review sorted by microecological regulators type.

Study	Design	Formulation	Population	Intervention		Control		Outcome	Outcome Measurement
				Type	N	Type	N		
**Prebiotics: n = 3**									
Alavi et al., 2010 [38]	Double-blind RCT	Ambrotose Complex (AC)	69	AC 1.3 g/day	33	Placebo	36	DAS28, patient global score, physician global score, SJS, TJS, ESR, CRP, ACPA, RF	6 months
Javadi et al., 2016 [37]	Double-blind RCT	Quercetin	50	Quercetin	25	Placebo	25	DAS28-ESR, PGA, TJC, SJC, early morning stiffness, VAS pain, HAQ, ESR, TNF-α	8 weeks
Khojah et al., 2018 [36]	open-label RCT	Resveratrol (RSV)	100	RSV 1g/day	50	Placebo	50	DAS28-ESR, moderate EULAR response, TJC, SJC, ESR, CRP, RF, TNF-α, IL-6	3 months
**Probiotics: n = 6**									
Hatakka et al., 2003 [42]	Double-blind RCT	*Lactobacillus casei* 01	21	≥108 CFU/capsule, daily	8	Placebo	13	DAS28-CRP, TJC, SJC, GH score, VAS, moderate EULAR response, hs-CRP, IL-1β, IL-6, IL-12, TNF-α, IL-10	1 year
Mandel et al., 2010 [41]	Double-blind RCT	*Bacillus coagulans* GBI-30, 6086	45	2 × 109 CFU/caplet, one caplet daily	23	Placebo	22	ACR20 response, SJC, TJC, HAQ core, VAS pain, VAS activity, ESR, CRP	8 weeks
Pineda et al., 2011 [40]	Double-blind RCT	*L. rhamnosus* GR-1 *L. reuteri* RC-14	29	2 × 109 CFU/capsule, one capsule twice daily	15	Placebo	14	ACR 20 response, DAS28-CRP, SJC, TJC, MS, HAQ score, VAS pain, VAS fatigue ESR, CRP, IL-1β, IL-1α IL-6, IL-8, TNF-α, IL-12p70, IL-15, IL-17 IL-10, GM-CSF	3 months
Alipour et al., 2014 [44]	Double-blind RCT	*Lactobacillus casei* 01	46	≥108 CFU/capsule, daily	22	Placebo	24	DAS28-CRP, TJC, S JC, GH score, VAS, moderate EULAR response, hs-CRP, IL-1β, IL-6, IL-12, TNF-α, IL-10	8 weeks
Zamani et al., 2016 [39]	Double-blind RCT	*Lactobacillus acidophilus* *Lactobacillus casei* *Bifidobacterium bifidum*	60	2×109 CFU/g(capsule) of each strain, one capsule daily	30	Placebo	30	DAS28-CRP, TJC, SJC, GH score, VAS, hs-CRP, insulin resistance, lipid concentrations, biomarkers and oxidative stress	8 weeks
Cannarella et al., 2021 [43]	Double-blind RCT	*Lactobacillus acidophilus* LA-14 *Lactobacillus casei* LC-11 *Lactococcus lactis* LL-23 *Bifidobacterium lactis* BL-04 *Bifidobacterium bifidum* BB-06	42	109 CFU/g of each strain, one sachet (2 g) daily	21	Placebo	21	DAS28-ESR, TJC, SJC, GH score, hs-CRP, ESR, ferritin, IL-6, TNF-α, IL-10, Oxidative and Nitrosative Stress Biomarkers	8 weeks
**Synbiotics: n = 3**									
Nenonen et al., 1998 [35]	Single-blind RCT	Probiotic: *L. plantarum* and *L. brevis* Prebiotic: Uncooked vegan diet (high dietary fibers)	39	Lactobacilli 2.4–4.5 × 1010 CFU/day in fermented wheat drink uncooked vegan diet	19	Normal diet	20	DAS28-ESR, SJC, TJC, HAQ, MS, VAS pain, CRP, ESR	3 months
Zamani et al., 2017 [45]	Double-blind RCT	Probiotic: *L. acidophilus*, *L. casei* *Bifidobacterium bifidum* Prebiotic: inulin	54	one synbiotic capsule (*Lactobacillus acidophilus*, *Lactobacillus casei* and *Bifidobacterium bifidum* (2 × 109 CFU/g each) and inulin 800 mg)/day	27	Placebo	27	DAS28-CRP, SJC, TJC, VAS pain, hs-CRP	2 months
Vadell et al., 2020 [46]	Single-blind crossover RCT	Probiotic: *L. plantarum* 299 v Prebiotic: Anti-inflammatory Diet rich in fatty acids and fibers	50	Probiotic shot: one shot 5 days a week Anti-inflammatory Diet	26	Typical Swedish diet	24	DAS28-CRP, DAS28-ESR, SJC, TJC, GH score ESR,	10 weeks

AC: Ambrotose complex; ACPA: anticitrullinated protein/peptide antibodies; ACR: American College of Rheumatology; CRP: C-reactive protein; CFU: colony-forming unit; hs-CRP: high-sensitivity C-reactive protein; DAS28: disease activity score in 28 joints; ESR: erythrocyte sedimentation rate; EULAR: European League Against Rheumatism; GH: global health; GM-CSF: granulocyte macrophage colony-stimulating factor; HAQ: health assessment questionnaire; L.: lactobacillus; MS: morning stiffness; NR: not reported; PGA: physician global assessment; RCT: randomized controlled trial; RF: rheumatoid factor; SJC: swollen joint count; SJS: swollen joint score; TJC: tender joint count; TJS: tender joint score; VAS: visual analogic scale.

## Data Availability

Not applicable.

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
