# Peer review of "Effects of Microecological Regulators on Rheumatoid Arthritis: A Systematic Review and Meta-Analysis of Randomized, Controlled Trials"

_nutrients, 2023, doi:10.3390/nu15051102_

Round 1

Reviewer 1 Report

The manuscript systematically reviews the implication of intestinal microecological regulators in the management of rheumatoid arthritis. The topic is relevant, and has potential, but the manuscript is poorly presented in its current form. Instructions for authors must be checked and applied (they are not optional but must be applied). Therefore, I recommend extensive improvements at the levels of literature search, editing, and structure, which are presented below:

1. The abstract is well structured but should not include headings. Please check the Instructions for authors and modify accordingly.

2. It is important to be consistent in denotation. Differences have been identified in terms of the databases evaluated (L20-21 Scopus is not listed in the abstract, but it is listed in L75-76 under search strategy etc.), and the reviewers who screened and assessed the quality of the studies (L22- 2 independent reviewers; L85- 3 independent reviewers).

3. In order to understand the role and advantages of adjuvant therapies in RA, it is important to briefly present the current allopathic therapies included in the pharmacotherapeutic management of RA (csDMARDs, bDMARDs, tsDMARDs) and what unmet needs these therapies present that could be addressed by microecological regulators. I suggest checking and referring to: PMID: 36058148.

4. RA presents an extremely complex pathophysiology that is not fully understood so far, so a brief presentation of the other risk factors that may influence the development and biomolecular mechanisms in the pathology besides microbiota imbalance would be necessary, as there may be cases where more than one risk factor is involved and an individual analysis of each factor should be made. I suggest checking and referring to: PMID: 34831081.

5. Bibliographic indexes represent the text itself and should not be linked to the previous word. Furthermore, MDPI style displays multiple consecutive indexes as [5-7], not [5; 6; 7]. You should revise the entire manuscript in terms of inserted references, position in the text, and the condition of never inserting them after punctuation marks. Please add/insert the references according to the Instructions for authors. References should be written in the MDPI style, which is provided by informatics software (EndNote®, Mendeley®, Zotero® etc.).

6. The Introduction should also include data related to the mechanism of action of microecological regulators in RA to understand their potential adjuvant role in managing the pathophysiological mechanisms of RA. I suggest checking and referring to: PMID: 34684377.

7. In L51 the definition of probiotics is given word for word and marked by quotation marks. It is advisable not to apply such techniques except in extraordinary cases when it is a generally accepted formula/theory/concept/definition that cannot have any other form. In this case, paraphrasing can be used.

8. Once an abbreviation has been established and explained, it will be used throughout the entire manuscript, with the exception of the abstract where it must be treated separately (i.e. TJC for tender joint count-L61 for the first time in the main text, not needed the explicit form in L91 etc.) throughout the entire manuscript. Moreover, several abbreviations (e.g., GC, NSAID, L110 etc.) have been included without explaining the explicit form. Please review the whole manuscript in terms of abbreviations.

9. Please revise the spelling of the most relevant terms in the entire manuscript, especially the keywords (synbiotics not synbioticss).

10. The aim of the study should be developed in a clearer and more relevant way because the scientific literature has already provided systematic reviews/meta-analyses of RCTs assessing the efficacy of pro/pre/synbiotics in RA. Therefore, answering the following question will help provide a relevant aim for the study: What is the novel or unique/special aspect that your review brings to the field? What is the reason for choosing the topic?

11. L71 requires the bibliographic resource of the accessed database (Prospero International Prospective Register of Systematic Reviews) (reference type web page, date of access, link, etc.).

12. Mesh is a controlled vocabulary tool that belongs only to MEDLINE and PubMed, so the search algorithm is only relevant for searching in PubMed. Please review the search strategy (L79) so that it is a valid search for all the databases mentioned in the strategy. Moreover, some search terms are written with a capital letter, others with a small letter. Be consistent in denotation.

13. Figure 1 (PRISMA diagram) contains several errors that need to be revised: spelling check in the text in the figure (sources instead of sources etc.); presentation in the diagram of the elements that led to the exclusion of the 2332 records during the screening period; the box containing full text-articles excluded, with reasons should be marked by an arrow and contain more than 10 (n=11), and for the last study not included in the meta-analysis, reasons should be given for its exclusion. The figure should be correlated with the written text, with the numerical data and it should be as clear as possible. Furthermore, the Title of the figure should not be included in the figure, but in the main text below the figure, following the Instructions for authors regarding figures. Please review all the figures in the manuscript. Also, ALL the figures must be provided in the clearest form, not BLURRED!!! Add clear figures!

14. Please check the numbering of chapters and subchapters, because errors have been identified (Study characteristics is chapter 4, not 2, etc.).

15. Tables 1 and 2 are actually figures, so they should be modified in editable format and modified following the characteristics of a table suggested by the Instructions for authors; the first column should become the last and contain the bibliographic references in square brackets which are automatically inserted and respect the MDPI style. The first column of the new table should include the name of the study. Any unexplained abbreviation in the table should be explained as a legend immediately after the table.

16. The name of the study should be included in Figure 2, not the main author presented as a bibliographic resource/study ID. The results of the assessment of risk of bias should be explained and interpreted in the discussion section.

17. It would be appropriate to present in the discussion section which are the most effective probiotics, prebiotics, synbiotics following RCTs evaluated, and which is the combination with the greatest potential as adjuvant therapy in RA. Please remove L314-315 as this is not the first study with this type of design, there are several conducted, please review the literature, and insist on the new contributions your paper can bring to the literature.

18. It is recommended to create a Conclusion section presenting the main data and contributions of this article and how the limitations presented in the discussion section can be addressed in future research directions.

Reviewer 2 Report

The submitted manuscript is a systematic review of experimental studies about the efficacy of probiotics (alone and/or with prebiotics) in RA patients.

The authors stated that the  study has been conducted according to  PRISMA guidelines, but the PRISMA module is not available. Please submit it.

Introduction and discussion about the role of microbiota and microbiota modulators must be improved. Feel free to read and discuss the main papers suggested below.

A conclusion section must be added.

References must be reviewed according to the journal style.

English language and grammarmust be revised, also for typo.

Suggested readings:

Alpizar-Rodriguez D, et al. doi: 10.1136/annrheumdis-2018-21451.

Elsouri K, et al. doi: 10.7759/cureus.15543

Topi S. et al. doi:10.3390/pathophysiology29030041

Lee JY, et al. doi: 10.3390/genes10100748

Santacroce L, et al. doi: 10.52586/4930

de Oliveira GLV, et al. doi: 10.1111/imm.12765.

Round 2

Reviewer 1 Report

The authors improved their manuscript.

Reviewer 2 Report

The current version of the manuscript, including supplementary files, has been improved in respect of the previous one and according the reviewers' suggestions.

However, all bacterial names must be italicized in text and tables.